# Vitamin D Status, Cardiovascular Risk Profile, and miRNA-21 Levels in Hypertensive Patients: Results of the HYPODD Study

**DOI:** 10.3390/nu14132683

**Published:** 2022-06-28

**Authors:** Domenico Rendina, Lanfranco D′Elia, Veronica Abate, Andrea Rebellato, Ilaria Buondonno, Mariangela Succoio, Fabio Martinelli, Riccardo Muscariello, Gianpaolo De Filippo, Patrizia D′Amelio, Francesco Fallo, Pasquale Strazzullo, Raffaella Faraonio

**Affiliations:** 1Department of Clinical Medicine and Surgery, Federico II University, 80131 Naples, Italy; domenico.rendina@unina.it (D.R.); lanfranco.delia@unina.it (L.D.); veronica.abate@unina.it (V.A.); drmuscariello@gmail.com (R.M.); strazzul@unina.it (P.S.); 2Department of Medicine, Clinica Medica 3, University of Padova, 35122 Padova, Italy; andrea.rebellato@aulss8.veneto.it (A.R.); francesco.fallo@unipd.it (F.F.); 3Department of Medical Science, Geriatric and Bone Diseases Unit, University of Turin, 10124 Torino, Italy; ilaria.buondonno@unito.it (I.B.); patrizia.damelio@chuv.ch (P.D.); 4Department of Molecular Medicine and Medical Biotechnology, Federico II University, 80131 Naples, Italy; mariangela.succoio@unina.it (M.S.); biotech88@hotmail.it (F.M.); 5Assistance Publique—Hôpitaux de Paris, Hôpital Robert Debré, Service d’Endocrinologie et Diabétologie Pédiatrique, 75015 Paris, France; gianpaolo.defilippo@aphp.fr; 6Department of Internal Medicine, Service of Geriatric Medicine and Geriatric Rehabilitation, University of Lausanne Hospital Centre, 1011 Lausanne, Switzerland

**Keywords:** hypertension, miR-21, hypovitaminosis D, cholecalciferol, calcitriol, clinical trial, inflammation

## Abstract

The vitamin D and microRNA (miR) systems may play a role in the pathogenesis of cardiometabolic disorders, including hypertension. The HYPODD study was a double-blind placebo-controlled trial aiming to assess the effects of cholecalciferol treatment in patients with well-controlled hypertension and hypovitaminosis D (25OHD levels < 50 nmol/L). In addition to this clinical trial, we also evaluated the effects of cholecalciferol and calcitriol treatment on miR-21 expression in vivo and in vitro, respectively. Changes in the cardiovascular risk profiles were evaluated in HYPODD patients treated with cholecalciferol (C-cohort) or with placebo (P-cohort). The miR-21circulating levels were measured in four C-cohort patients and five P-cohort patients. In vitro, the miR-21 levels were measured in HEK-293 cells treated with calcitriol or with ethanol vehicle control. Cholecalciferol treatment increased 25OHD levels and reduced parathormone, total cholesterol, and low-density lipoprotein cholesterol levels in C-cohort patients, whereas no significant changes in these parameters were observed in P-cohort patients. The miR-21 circulating levels did not change in the C- or the P-cohort patients upon treatment. Calcitriol treatment did not affect miR-21 levels in HEK-293 cells. In conclusion, hypovitaminosis D correction ameliorated the cardiovascular risk profiles in hypertensive patients treated with cholecalciferol but did not influence the miR-21 expression.

## 1. Introduction

Hypertension is an important public health challenge because of its high frequency and concomitant risks of cardiovascular and kidney diseases [1,2]. It is recognized as the leading risk factor for mortality, and it is ranked third as a cause of disability-adjusted life–years [3,4]. Experimental and epidemiological studies suggest that the vitamin D biological system may play a role in the pathogenesis of hypertension, hypertension-related organ damage, and, in general, cardiovascular diseases [5,6,7]. Vitamin D deficiency, defined as 25OHD serum levels <50 nmol/L [8], has been associated with an increased risk of hypertension, hypertension-related organ damage, heart failure, and coronary artery disease [5,6,7,8]. A recent meta-analysis of randomized controlled trials indicated that the correction of vitamin D deficiency improves arterial stiffness, a factor in turn associated with systolic hypertension, coronary artery disease, stroke, and heart failure [9]. In this scenario, different experimental models of disease featured a relationship between vitamin D status and microRNA (miR) expression [10,11,12]. The miRs are short, endogenous, single-strand, noncoding RNA molecules responsible for the post-transcriptional control of gene expression by interacting with target messenger RNA at specific sites to induce the cleavage of the message or to inhibit its translation [13]. The miRs are considered critical modulators of gene expression in many physiological and pathological processes, including hypertension and hypertension-related organ damage [14,15]. In particular, experimental studies suggested the possible involvement of miR-21 in the pathogenesis of essential hypertension and the occurrence of a complex bidirectional relationship between miR-21 and the vitamin D biological system in various physio-pathological conditions [15,16,17,18,19,20,21]. For example, the role of vitamin D in balancing miR-21 expression was shown in atherosclerosis and in osteoporosis [21,22], and a significant increase in miR-21 expression was described after 1,25(OH)_2_D_3_ treatment in Wistar rats with kidney ischemia–reperfusion injury [23]. Moreover, miR-21 plays a dynamic role in the inflammatory response [24]. The association between vitamin D deficiency and inflammatory status has been well established [25], and hypertension is associated with increased serum levels of metabolic and inflammatory markers [26]. Taking these data into account altogether, it is reasonable to study the relationship of vitamin D to miR-21 in patients with hypertension. The HYPODD (HYPOvitaminosis D and organ Damage) study was conceived against this background: As previously described, it was a no-profit multicenter parallel-group double-blind placebo-controlled randomized trial aiming to assess the effects of cholecalciferol supplementation on the cardiovascular risk profiles of patients with essential hypertension and vitamin D deficiency [27]. To assess miR expression, RNA samples were obtained from participating patients. In this article, we report: (a) the changes in cardiovascular risk profiles we observed in the HYPODD participants upon the correction of hypovitaminosis D; (b) the circulating miR-21 levels before and after treatment with placebo or cholecalciferol; and (c) the in vitro effects of 1,25(OH)_2_D_3_ on miR-21 intracellular expression levels.

## 2. Materials and Methods

### 2.1. The HYPODD Study

The HYPODD study was conceived by the Working Group on Vitamin D and Cardiorenal Disorders, established jointly by the Italian Society of Hypertension (SIIA) and the Forum in Bone and Mineral Research (FBMR) and coordinated by the Federico II University of Naples European Society of Hypertension Excellence Center [27]. Initially, twelve SIIA and FBMR Centers joined HYPODD, but nine of them found it difficult to enroll the participants according to the trial protocol and dropped out of the project. This manuscript thus refers to the patients enrolled at three centers (Naples, Turin, and Padua). All the patients had idiopathic hypertension under optimal pharmacological control and had 25OHD serum levels <50 nmol/L. Based on a double-blind randomization schedule, the patients were allocated to the cholecalciferol cohort (C-cohort) or to a placebo cohort (P-cohort). The C-cohort patients were instructed to orally take 50,000 UI (1.25 mg) of cholecalciferol every week for 8 weeks and subsequently 50,000 UI of the same substance every month for 10 months. The P-cohort patients were instructed to orally take the placebo preparation, which had identical organoleptic characteristics as the cholecalciferol preparation, at the same time points. Both the active drug and the identical placebo preparation were produced by the ABIOGEN drug company, which also handled the randomization procedure using a computer-generated allocation schedule. The study investigators remained blinded to the patient allocation until the completion of the statistical analyses. Throughout the study, systolic and diastolic pressure were maintained respectively below 140 mmHg and 90 mmHg by appropriate modifications of the pharmacological therapy according to a predetermined schedule [27]. The main endpoints of the study were the assessment of the effects of cholecalciferol supplementation on the consumption of antihypertensive drugs needed to maintain optimal blood pressure control [1] and of cholecalciferol supplementation on the progression of organ damage. To reach these primary endpoints, we had estimated a sample size of 240 patients. The HYPODD study protocol was approved by the Carlo Romano Ethical Committee of Federico II University of Naples Medical School (prot.41/12) and was registered at the Agenzia Italiana del Farmaco-Osservatorio sulla Sperimentazione Clinica del Farmaco (AIFAOsSC) and the EUDRACT sites (n_2012-003514-14). Written informed consent was obtained from each patient enrolled in the study.

### 2.2. miR-21 In Vivo Study

The miR-21 circulating levels were measured at enrollment (T0) and after 2 months (T2) in the first 9 patients enrolled in the HYPODD study. For the measurement of miR-21 levels, blood samples were collected into Tempus™ blood RNA tubes (Thermo Fisher Scientific, Life Technologies Corporation, Austin, TX, USA) and subsequently stored upon freezing at −20 °C. Total RNA was prepared from archived samples using a MagMAX™ for Stabilized Blood Tubes RNA Isolation Kit, according to the manufacturer’s instructions. Briefly, thawed samples were transferred into conical cubes and mixed with 3 mL of Tempus^®^ 1XPBS. Pellets obtained after centrifugation at 5000× *g* for 15 min at 4 °C were treated with Tempus^®^ Pre-Digestion Wash. After a second centrifugation at 5000× *g* for 10 min at 4 °C, pellets were mixed with 120 µL of a mixture containing Tempus Resuspension Solution and Tempus Proteinase; then the mixture was vortexed, and 10 µL of TURBO™ DNase was added. RNAs were then bound to RNA Binding Beads and magnetically captured for 3 min. This operation was repeated twice for 2 min after 2 different washes of the tubes. The beads were dried and mixed with 40 µL of elution buffer and magnetically captured 1 last time for 3 min. The concentration of extracted RNAs was assessed by spectrophotometry using NanoDrop (Thermo 205 Scientific, Wilmington, DE, USA), and RNA purity was estimated by measuring the OD 260/280 ratios. The cDNA templates for the evaluation of mature miR levels were prepared from input RNAs (10 ng) using a TaqMan™ Advanced miR cDNA Synthesis Kit (Thermo Fisher Scientific, Richardson, TX, USA, Cat. No. A28007) following the manufacturer’s protocol. Real-time quantitative PCR (RT-qPCR) assays were performed in triplicate on diluted cDNA templates (1:10) using the TaqMan^®^ Advanced miRNA Assay for hsa-miR-21-5p (Assay ID: 477975) and for hsa-miR-16-5p (Assay ID 477860) (Thermo Fisher Scientific), this latter used as endogenous reference miR. The relative quantification of miR expression levels was performed according to the 2^−ΔΔCt^ method [28]. Quantitative results were obtained using miR-16-5p as a reference miR for normalization, previously described as a suitable normalizer miR in blood [29].

### 2.3. miR-21 In Vitro Study

Human embryonic kidney 293 cells (HEK-293) were grown in Dulbecco’s modified minimal medium (DMEM) supplemented with 10% FBS in 5% CO2 atmosphere at 37 °C. For the 1,25(OH)_2_D_3_ treatments, cells were plated at a density of 5 × 10^5^ in 60 mm plates, allowed to grow overnight, and then treated for 24 h with ethanol vehicle control or with 1,25(OH)_2_D_3_ (Sigma-Aldrich, Sheboygan Falls, WI, USA, S.r.l. D1530) at final concentration of 1 nM, 10 nM, or 100 nM.

Total RNA from HEK293 cells treated with 1,25(OH)_2_D_3_ was prepared using TRIzol Reagent (Thermo Fisher Scientific) and quantified spectrophotometrically by Nanodrop (Thermo Scientific). To evaluate the levels of mature miR-21-5p upon 1,25(OH)_2_D_3_ treatments, the cDNA templates on RNAs from the treated cells and the relative quantification were performed as previously described [30] using TaqMan™. Advanced miR cDNA Synthesis Kit and miRNA assays were utilized for the evaluation of miRNA levels from blood samples (see above).

To measure the coding mRNA levels in the 1,25(OH)_2_D_3_-treated cells, the first-strand cDNAs were produced from one µg of total RNA using a SensiFAST cDNA Synthesis Kit (Bio-Line, Aurogene, Italy) according to the manufacturer’s instructions. RT-qPCR was carried out on an iCycler (BioRad) using SensiFAST SYBR No-ROX (Bio-Line). The housekeeping c-ABL mRNA was used for internal normalization. PCR reactions were performed on biological triplicates and in experimental triplicates. Fold changes were calculated using the 2^−ΔΔCt^ method [26]. The sequences of the primer pairs used were: c-Abl, 5′-TGGAGATAACACTCTAAGCATAACTAAAGGT-3′ and 5′-GATGTAGTTGCTTGGGACCCA-3′; VDR, 5′-ATAAGACCTACGACCCCACCTA-3′, and 5′-GGACGAGTCCATCATGTCTGAA-3′; CYP24A1, 5′-GCACAAGAGCCTCAACACCAA-3′, and 5′-AGA CTGTTTGCTGTCGTTTCCA-3′.

### 2.4. Statistical Analysis

All statistical analyses were performed using the SPSS software, version 25 (SPSS Inc., Chicago, IL, USA). The distribution of variables was assessed with the Kolmogorov–Smirnov test (*p* > 0.05). Data were expressed as mean ± standard deviation and absolute (percentage) number for qualitative and quantitative data, respectively. For the HYPODD study, the analysis of variance and the contingency table X^2^ test were used to compare the qualitative and quantitative data, respectively. Differences from baseline after treatment in the C- and P-cohorts were examined using Student *t*-tests for paired samples. The relationships between the quantitative data in the different study cohorts were tested by performing linear regression analysis. Paired-sample *t* tests were used to compare baseline with post-treatment miR levels in individual participants. Independent-sample *t* tests were used to compare between-group differences. Changes in the levels of miRs were calculated as final minus basal values. Analysis of variance (ANOVA) was used to assess differences in the miR changes between the two groups. Two-sided *p* values below 0.05 were considered statistically significant.

## 3. Results

### 3.1. The HYPODD Study

The baseline (T0) anthropometric and clinical data of the patients allocated to the active treatment with cholecalciferol (C-cohort) and those allocated to treatment with placebo (P-cohort) are shown in Table 1.

No significant differences were observed in the clinical and anthropometric parameters evaluated at T0 between the C- and P-cohorts.

The clinical and biochemical parameters measured during the study are reported in Table 2.

The changes in clinical and biochemical parameters observed during the study, expressed as difference from T0 (Δ), are reported in Table 3.

As reported in Table 2, no significant differences were observed in biochemical parameters measured at enrollment in the two study cohorts. Cholecalciferol treatment significantly increased 25OHD levels during the study, correcting vitamin D deficiency in all patients in the C-cohort two months after the enrollment (T2), six months after the enrollment (T6), and twelve months after the enrollment (T12). No significant changes were observed in serum 25OHD levels in the P-cohort. In C-cohort patients, we observed a significant reduction in the serum levels of intact parathormone (PTH), total cholesterol (T-Chol), low-density lipoprotein (LDL)—Chol, and triglycerides (Tri) at T2, T6, and T12 (*p* < 0.05 in all cases; Table 2). No significant change in any biochemical parameter measured was observed in the patients receiving placebo treatment.

These results were also confirmed after we analyzed the differences (Δ) from baseline (T0) for each biochemical parameter, as reported in Table 3.

During the study follow-up, 8 C-cohort patients and 10 P-cohort patients did not change their antihypertensive therapy, 6 C-cohort patients and 7 P-cohort patients reduced the dosage and/or the number of antihypertensive drugs taken, and 1 C-cohort patient and 3 P-cohort patients increased the dosage and/or the number of antihypertensive drugs taken. As reported in Table 3, at T12, the Δ in systolic blood pressure measured in C-cohort patients is significantly higher than that measured in P-cohort study.

### 3.2. miR-21 In Vivo Study

The miR-21 circulating levels were measured in the first 9 patients enrolled in the HYPODD study (male (M):female (F) 8:1; mean age 60.0 ± 4.9 years (range 51–66 years); body mass index (BMI) 27.2 ± 2.4 kg/m^2^ (range 23.3–29.9); 8 treated with multiple antihypertensive drugs; 1 treated with a single antihypertensive drug). A total of 4 patients were enrolled in the C-cohort and received cholecalciferol treatment (M:F 4:0; mean age 60.3 ± 6.8 years (range 51–66 years); BMI 28.0 ± 0.8 kg/m^2^ (range 27.3–29.1)), and 5 patients were enrolled in the P-cohort and received placebo treatment (M:F 4:1; mean age 59.8 ± 3.8 years (range 54–63 years); BMI 26.5 ± 3.1 kg/m^2^ (range 23.3–29.9)). No significant differences were observed in the anthropometric parameters between the two subgroups at T0. The circulating levels of 25OHD and miR-21, measured at T0 and T2 in both study cohorts, are shown in Figure 1.

At T0, the circulating levels of miR-21 (6.9 ± 0.8 vs. 6.6 ± 0.3, miR-21ΔCT, for C- and P-cohort, respectively) and 25OHD (39.3 ± 8.4 vs. 44.6 ± 11.7, nmol/L, for C- and P-cohort, respectively) were similar (in both cases *p* > 0.05, analysis of variance). At T2, the 25OHD serum levels significantly increased in patients receiving cholecalciferol therapy (39.3 ± 8.4 vs. 77.1 ± 17.8, nmol/L, *p* < 0.01, T-test for paired samples). No change in 25OHD serum levels was seen in patients receiving placebo (44.6 ± 11.7 vs. 54.5 ± 11.6, nmol/L, *p* > 0.05, T-test for paired samples). The circulating miR-21 levels did not significantly change at T2 either in C-cohort (6.9 ± 0.8 vs. 7.4 ± 0.7, miR-21ΔCT, at T0 and T2, respectively; *p* > 0.05, T-test for paired samples) or in P-cohort (6.6 ± 0.3 vs. 7.1 ± 0.5, miR-21ΔCT, at T0 and T2, respectively; *p* > 0.05, T-test for paired samples) patients. None of the patients changed his/her antihypertensive treatment during the study.

### 3.3. miR-21 In Vitro Study

We also sought to establish the direct effect of 1,25(OH)_2_D_3_, the most active biological form of vitamin D, on miR-21 expression in vitro.

As shown in Figure 2a, in HEK-293 cells, the miR-21 levels were unaffected by treatments with different concentrations of 1,25(OH)_2_D_3_, resulting in the absence of miR-21 changes compared with cells incubated with vehicle alone. As a positive control, we also tested the cellular response to 1,25(OH)_2_D_3_ by measuring the expression patterns of selected genes related to 1,25(OH)_2_D_3_ signaling, including vitamin D receptor (VDR) and cytochrome P450 family 24 subfamily A member 1 (CYP24A1) [31]. The data confirmed a strong transcriptional induction of CYP24A1, a VDR target gene (Figure 2b), thus further supporting the conclusion that vitamin D treatment does not directly affect miR-21 expression either in vivo or in vitro.

## 4. Discussion

In this study, oral cholecalciferol treatment was very effective for the correction of vitamin D deficiency and the maintenance of euvitaminosis D in patients with pharmacologically well-controlled essential hypertension. The increase in 25OHD serum levels was associated with a significant improvement in the cardiovascular risk profile, since cholecalciferol treatment significantly reduced the serum levels of T-Chol, LDL-Chol, Tri, and PTH and caused a significant reduction in systolic blood pressure values.

The reduction of T-Chol and LDL-Chol levels observed in our study is similar to that obtained using a nutraceutical combination containing berberine, policosanol, and red yeast rice, drugs regularly used in the clinical management of patients with dyslipidemia [32]. In addition, the reduction in atherogenic lipids is similar to this observed in previous studies performed in adults and in children [33,34,35,36,37], but, of interest, none of these studies was performed in adult hypertensive patients, who are subjects at increased cardiovascular risk.

With regard to PTH and to the elevated PTH levels observed in our vitamin D-deficient patients, several studies support the detrimental effects of the secondary hyperparathyroidism induced by vitamin D deficiency on heart function, leading to cardiac hypertrophy, altered cardiac remodeling, arrhythmias, and sudden death [38,39,40]. The significant reduction in the PTH levels obtained as a result of the correction of hypovitaminosis D marks a meaningful improvement in the patient’s cardiovascular risk profile.

The correction of vitamin D deficiency on the other hand did not influence the circulating miR-21 levels in our study, nor did we detect any association between miR-21 changes and cardiovascular or biochemical parameters: These results are at variance with previous reports suggesting that miR-21 regulation associated with vitamin D deficiency might play a significant role in the pathogenesis of chronic metabolic and inflammatory conditions such as hypertension, disorders of lipid metabolism, and periodontitis as well as type 2 diabetes mellitus and other glucometabolic disorders [5,6,7,17,18,19,20,21,22,23,41,42,43,44,45]. Our parallel in vitro studies supported the in vivo results in that HEK-293 cell miR-21 levels were unaffected by treatment with different concentrations of 1,25(OH)_2_D_3_. Our results thus indicate that the improvements in the cardiovascular risk profiles of hypertensive patients was not linked to changes in miR-21 expression.

Our study had of course an important limitation in its small sample size, which precluded the possibility of meeting the primary endpoint of the study, which was the reduction of the consumption of antihypertensive drugs because of better blood pressure control possibly associated with the correction of vitamin D deficiency. Indeed, a clinically meaningful decrease in systolic blood pressure in the patient group receiving cholecalciferol was observed, but this result must be considered with great caution considering the small number of patients enrolled and the changes in the antihypertensive treatment that were adopted by the clinicians during the study follow-up according to the predetermined flow chart [27]. On the other hand, even with the small number of patients enrolled, the overall improvement in the cardiovascular risk profile linked to the correction of hypovitaminosis D was apparent and statistically significant, definitely warranting attention through further clinical studies. The study results may contribute to ameliorating the clinical management of patients with hypertension when it is also well controlled by pharmacological treatment. Our study demonstrates that the correction of hypovitaminosis D improved cardiovascular risk profiles, strongly indicating the need for further studies of ad hoc design. In addition, the study results provide support for measuring 25OHD serum levels in patients with cardiovascular disorders [46,47]. Finally, the circulating miR-21 levels were not evaluated at T6 and T12, and thus, it is not possible to rule out that their variations could have influenced the biological effects on the cardiovascular risk profiles we observed at T6 and T12. The present work did, however, also have some important strengths, such as the adoption of a double-blind placebo-controlled design [48]. Furthermore, the quantification of miR-21 levels was performed using RT-qPCR, the gold standard method for measuring miRNA profiles, characterized by high reproducibility and specificity [28,29]. On the other hand, it is not excluded that miRs other than miR-21 may have peculiar patterns related to circulating levels of vitamin D; therefore, studies with the same assumption are necessary that take into consideration full miRNoma analysis.

## 5. Conclusions

In conclusion, even with a small sample size, the HYPODD study results demonstrated that the correction of hypovitaminosis D ameliorates the cardiovascular risk profiles of patients with essential hypertension. The concomitant assessment of the possible changes in miR-21 expression in vivo and in vitro did not support the hypothesis that miR-21 regulation has a causal role in this regard.

## Figures and Tables

**Figure 1 nutrients-14-02683-f001:**
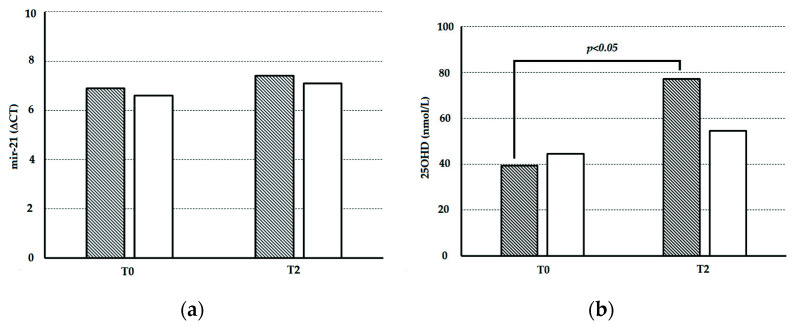
Circulating levels of miR-21 and 25OHD in the study cohorts before and after treatment with cholecalciferol or placebo: (**a**) miR-21 circulating levels in C-cohort HYPODD patients (dark histograms) and in P-cohort HYPODD patients (white histograms) before (T0) and two months (T2) after treatment with cholecalciferol (C-cohort) or with placebo (P-cohort); (**b**) 25OHD serum levels in C-cohort HYPODD patients (dark histograms) and in P-cohort HYPODD patients (white histograms) before (T0) and two months (T2) after treatment with cholecalciferol (C-cohort) or with placebo (P-cohort).

**Figure 2 nutrients-14-02683-f002:**
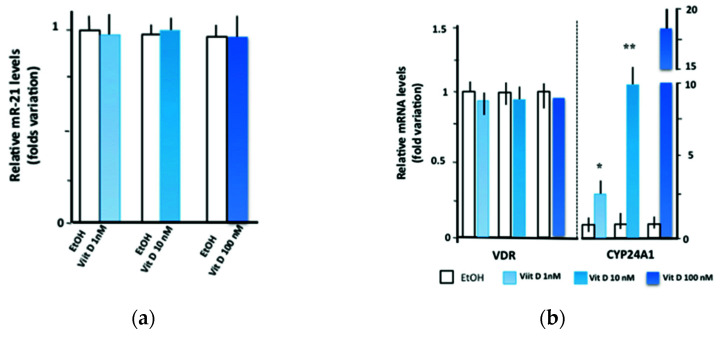
In vitro effects of 1,25(OH)_2_D_3_ treatment on miR-21, vitamin D receptor messenger RNA, and cytochrome P450 family 24 subfamily A member 1 messenger RNA expression levels. Human embryonic kidney (HEK293) cell lines were treated with different concentrations (1 nM, 10 nM, and 100 nM) of 1,25(OH)_2_D_3_ for 24 h. (**a**) Real-time quantitative polymerase chain reaction (RT-qPCR) analyses of miR-21 in HEK293 cells treated with 1,25(OH)_2_D_3_. The relative levels of miR-21 were evaluated using as control cells treated with ethanol vehicle control (EtOH) alone. MiR-16 was used as an internal standard for normalization. (**b**) Single gene expression analysis in HEK 293 cells treated with 1,25(OH)_2_D_3_. The relative amounts of vitamin D receptor (VDR) and cytochrome P450 family 24 subfamily A member 1 (CYP24A1) messenger ribonucleic acid (mRNAs) after 1,25(OH)_2_D_3_ treatments were evaluated with RT-qPCR using as control EtOH–treated cells. Statistical differences of ΔCt were marked with * standing for *p* < 0.05 and ** standing for *p* < 0.01.

**Table 1 nutrients-14-02683-t001:** Baseline anthropometric and clinical data of the study patients.

	C-Cohort	P-Cohort	*p*
**NumberN**	15	20	
**M:F**	11 (73.3):4 (26.7)	13 (65.0):7 (35.0)	*0.72*
**Age (years)**	59.3 ± 8.3	60.8 ± 6.4	*0.55*
**BMI (kg/m^2^)**	27.3 ± 3.3	27.4 ± 3.4	*0.94*
**Waist Circumference (cm)**	98.0 ± 6.3	95.3 ± 9.1	*0.27*
**Diuretics (no:yes)**	8 (53.3):7 (46.7)	8 (42.1):11 (57.9)	*0.73*
**Calcium antagonists (no:yes)**	9 (60.0):6 (40.0)	16 (84.2):3 (15.8)	*0.14*
**RAAS Inhibitors (no:yes)**	3 (20.0):12 (80.0)	3 (15.8):16 (84.2)	*0.98*
**Beta-blockers (no:yes)**	12 (80.0):3 (20.0)	13 (68.4):6 (31.6)	*0.69*
**Alfa-blockers (no:yes)**	13 (86.7):2 (13.3)	18 (94.7):1 (5.3)	*0.57*
**Number of hypertensive drugs used (*n*)**	2.0 ± 1.1	1.94 ± 1.1	*0.88*

Data are expressed as mean ± standard deviation or absolute number (percentage) for continuous or categorical variables, respectively. C-Cohort: patients with essential hypertension and vitamin D deficiency treated with cholecalciferol. P-Cohort: patients with essential hypertension and vitamin D deficiency treated with placebo. M: male. F: female. BMI: body mass index. The *p* values were estimated using ANOVA or X^2^ test for continuous or discrete variables, respectively; *p* value < 0.05 was considered statistically significant.

**Table 2 nutrients-14-02683-t002:** Clinical and biochemical parameters in the HYPODD study cohorts.

	C-Cohort				P-Cohort			
	T0	T2	T6	T12	T0	T2	T6	T12
**25OHD (nmol/L)**	42.6 ± 10.4	78.1 ± 17.8 ^A^	79.9 ± 14.3 ^A^	82.9 ± 11.1 ^A^	43.1 ± 10.1	54.5 ± 11.2 ^B^	50.3 ± 10.3 ^B^	50.2 ± 9.9 ^B^
**PTH (pmol/L)**	4.75 ± 2.07	4.39 ± 1.84 ^A^	4.21 ± 1.79 ^A^	4.20 ± 1.68 ^A^	4.68 ± 1.97	4.58 ± 1.76	4.61 ± 1.59	4.60 ± 1.87
**Glu (mmol/L)**	4.97 ± 0.51	4.81 ± 0.48	4.73 ± 0.49	4.68 ± 0.48	5.09 ± 0.52	5.12 ± 0.51	5.14 ± 0.51	5.14 ± 0.52
**Ins (pmol/L)**	41.0 ± 25.0	39.6 ± 22.2	36.8 ± 19.5	35.4 ± 19.5	49.3 ± 22.9	50.1 ± 21.5	49.4 ± 21.5	49.3 ± 18.8
**HOMA-IR**	1.21 ± 0.22	1.21 ± 0.24	1.28 ± 0.21	1.32 ± 0.23	1.03 ± 0.24	1.02 ± 0.21	1.04 ± 0.23	1.04 ± 0.25
**T-Chol (mmol/L)**	4.67 ± 0.77	4.23 ± 0.47 ^A^	4.36 ± 0.44 ^A^	4.37 ± 0.44 ^A^	5.03 ± 0.53	4.92 ± 0.46	4.82 ± 0.47	4.88 ± 0.44
**HDL-Chol (mmol/L)**	1.43 ± 0.39	1.40 ± 0.37	1.39 ± 0.36	1.38 ± 0.35	1.52 ± 0.50	1.52 ± 0.40	1.48 ± 0.42	1.50 ± 0.43
**LDL-Chol (mmol/L)**	2.72 ± 0.90	2.52 ± 0.75 ^A^	2.48 ± 0.78 ^A^	2.47 ± 0.74 ^A^	2.93 ± 0.55	2.75 ± 0.55	2.80 ± 0.68	2.79 ± 0.64
**Tri (mmol/L)**	1.35 ± 0.80	1.16 ± 0.58 ^A^	1.15 ± 0.51 ^A^	1.15 ± 0.55 ^A^	1.34 ± 0.70	1.31 ± 0.61	1.34 ± 0.57	1.31 ± 0.56
**Crea (µmol/L)**	78.7 ± 15.9	78.7 ± 15.9	76.0 ± 15.0	76.9 ± 11.5	71.6 ± 15.0	71.6 ± 15.0	75.1 ± 11.5	76.9 ± 15.9
**Ca (mmol/L)**	2.41 ± 0.13	2.41 ± 0.13	2.40 ± 0.12	2.41 ± 0.13	2.44 ± 0.14	2.44 ± 0.14	2.43 ± 0.12	2.43 ± 0.13
**Alb (g/L)**	47 ± 3	47 ± 4	45 ± 6	46 ± 4	47 ± 4	46 ± 5	46 ± 5	46 ± 5
**Mg (mmol/L)**	0.75 ± 0.23	0.76 ± 0.28	0.77 ± 0.26	0.79 ± 0.25	0.67 ± 0.25	0.75 ± 0.23	0.76 ± 0.25	0.76 ± 0.21
**P (mmol/L)**	1.07 ± 0.13	1.07 ± 0.13	1.13 ± 0.16	1.16 ± 0.16	1.16 ± 0.20	1.16 ± 0.20	1.10 ± 0.16	1.07 ± 0.17
**Na (mmol/L)**	141.4 ± 1.5	141.6 ± 1.6	141.2 ± 1.5	141.9 ± 1.8	141.8 ± 1.9	142.1 ± 1.9	141.8 ± 1.8	141.8 ± 1.8
**K (mmol/L)**	4.28 ± 0.46	4.27 ± 0.51	4.30 ± 0.51	4.28 ± 0.41	4.21 ± 0.49	4.26 ± 0.47	4.27 ± 0.48	4.28 ± 0.39
**TSH (mUI/L)**	2.37 ± 0.91	2.19 ± 0.88	1.87 ± 0.76	1.71 ± 0.67	1.90 ± 0.81	2.08 ± 0.89	2.16 ± 0.71	2.18 ± 0.61
**SBP (mmHg)**	133.8 ± 6.1	128.7 ± 5.9	127.4 ± 5.8	124.8 ± 6.1	132.0 ± 6.3	128.0 ± 6.0	129.5 ± 6.1	131.3 ± 5.8
**DBP (mmHg)**	80.7 ± 6.1	80.1 ± 5.6	79.8 ± 5.6	79.7 ± 6.0	79.3 ± 7.5	80.4 ± 7.1	82.1 ± 6.9	83.4 ± 7.0

Data are expressed as mean ± standard deviation. T0: baseline (enrollment). T2: 2 months from T0. T6: 6 months from T0. T12: 12 months from T0. C-Cohort: patients with essential hypertension and vitamin D deficiency treated with cholecalciferol. P-Cohort: patients with essential hypertension and vitamin D deficiency treated with placebo. 25OHD: calcifediol. PTH: intact parathormone. Glu: glucose. Ins: insulin. HOMA-IR: homeostatic model assessment for insulin resistance. T-Chol: total cholesterol. HDL-Chol: high-density lipoprotein cholesterol. LDL-Chol: low-density lipoprotein cholesterol. Tri: triglycerides. Crea: creatinine. Ca: calcium. Alb: albumin. Mg: magnesium. P: phosphate. Na: sodium. K: potassium. TSH: thyroid-stimulating hormone. SBP: systolic blood pressure. DBP: diastolic blood pressure. A = significantly different compared with T0; T-test for paired samples, a *p* value < 0.05 was considered statistically significant. B = significantly different compared with C-cohort; ANOVA *p* value < 0.05 was considered statistically significant.

**Table 3 nutrients-14-02683-t003:** Differences from baseline (T0) in the biochemical parameters measured at various time points during the study in the two patient cohorts (expressed as Δ).

	C-Cohort			P-Cohort		
	ΔT2	ΔT6	ΔT12	ΔT2	ΔT6	ΔT12
**25OHD (nmol/L)**	+35.5 ± 8.1 ^A^	+37.3 ± 8.3 ^A^	+40.3 ± 91.1 ^A^	+11.4 ± 3.2	+7.2 ± 4.1	+7.1 ± 3.9
**PTH (pmol/L)**	−0.36 ± 0.09 ^A^	−0.54 ± 0.09 ^A^	−0.55 ± 0.10 ^A^	−0.10 ± 0.03	−0.07 ± 0.02	−0.08 ± 0.03
**Glu (mmol/L)**	−0.16 ± 0.04	−0.24 ± 0.06	−0.29 ± 0.07	+0.03 ± 0.01	+0.05 ± 0.01	+0.05 ± 0.01
**Ins (pmol/L)**	−1.4 ± 0.7	−4.2 ± 1.3	−5.6 ± 1.5	+0.8 ± 0.3	+0.1 ± 0.1	+0.0 ± 0.1
**HOMA-IR**	0.0 ± 0.15	+0.07 ± 0.12	+0.09 ± 0.12	−0.01 ± 0.11	0.01 ± 0.09	0.01 ± 0.09
**T-Chol (mmol/L)**	−0.44 ± 0.09 ^A^	−0.31 ± 0.08 ^A^	−0.30 ± 0.09 ^A^	−0.11 ± 0.05	−0.21 ± 0.07	−0.15 ± 0.06
**HDL-Chol (mmol/L)**	−0.03 ± 0.01	−0.04 ± 0.02	−0.05 ± 0.02	+0.00 ± 0.01	−0.04 ± 0.02	−0.02 ± 0.02
**LDL-Chol (mmol/L)**	−0.20 ± 0.05	−0.24 ± 0.05 ^A^	−0.25 ± 0.04 ^A^	−0.18 ± 0.09	−0.13 ± 0.07	−0.14 ± 0.08
**Tri (mmol/L)**	−0.19 ± 0.07 ^A^	−0.20 ± 0.07 ^A^	−0.20 ± 0.06 ^A^	−0.03 ± 0.02	0.00 ± 0.02	−0.03 ± 0.02
**rea (µmol/L)**	0.0 ± 0.9	−2.7 ± 1.0	−1.8 ± 1.1	+0.0 ± 0.9	+3.5 ± 1.4	+5.3 ± 1.9
**Ca (mmol/L)**	0.00 ± 0.01	−0.01 ± 0.01	0.00 ± 0.01	+0.00 ± 0.01	−0.01 ± 0.01	−0.01 ± 0.01
**Alb (g/L)**	0 ± 2	−2 ± 1	−1 ± 1	−1 ± 1	−1 ± 1	−1 ± 1
**Mg(mmol/L)**	+0.01 ± 0.01	+0.02 ± 0.01	+0.04 ± 0.01	+0.08 ± 0.05	+0.09 ± 0.05	+0.09 ± 0.04
**P (mmol/L)**	+0.00 ± 0.03	+0.06 ± 0.04	+0.09 ± 0.06	0.00 ± 0.02	−0.06 ± 0.03	−0.09 ± 0.04
**Na (mmol/L)**	+0.2 ± 0.6	−0.2 ± 0.5	+0.5 ± 0.4	+0.3 ± 0.4	+0.0 ± 0.8	+0.0 ± 0.8
**K (mmol/L)**	−0.01 ± 0.01	+0.02 ± 0.01	+0.00 ± 0.03	+0.05 ± 0.04	+0.06 ± 0.04	+0.07 ± 0.04
**TSH (mUI/L)**	−0.18 ± 0.10	−0.50 ± 0.11	−0.66 ± 0.12	+0.18 ± 0.09	+0.26 ± 0.11	+0.28 ± 0.12
**SBP (mmHg)**	−5.1 ± 3.9	−6.4 ± 3.8	−9.0 ± 4.1 ^A^	−4.0 ± 3.1	−2.5 ± 3.1	−0.7 ± 2.8
**DBP (mmHg)**	−0.6 ± 2.6	−0.9 ± 2.1	−1.0 ± 1.9	+1.1 ± 2.1	+2.8 ± 1.9	+4.1 ± 2.4

Data are expressed as mean ± standard deviation. C-Cohort: patients with essential hypertension and vitamin D deficiency treated with cholecalciferol. P-Cohort: patients with essential hypertension and vitamin D deficiency treated with placebo. T = time. T0: value at T0 (enrollment). ΔT2: value measured at 2 months—value measured at T0. T6: value measured at 6 months—value measured at T0. T12: value measured at 12 months—value measured at T0. 25OHD: calcifediol. PTH: intact parathormone. Glu: glucose. Ins: insulin. HOMA-IR: homeostatic model assessment for insulin resistance. T-Chol: total cholesterol. HDL-Chol: high-density lipoprotein cholesterol. LDL-Chol: low-density lipoprotein cholesterol. Try: triglycerides. Crea: creatinine. Ca: calcium. Alb: albumin. Mg: magnesium. P: phosphate. Na: sodium. K: potassium. TSH: thyroid-stimulating hormone. SBP: systolic blood pressure. DBP: diastolic blood pressure. A = significantly different compared with P-cohort; ANOVA *p* value < 0.05 was considered statistically significant.

## Data Availability

All datasets generated during and/or analyzed during the current study are not publicly available but are available from the corresponding author on reasonable request.

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
