# Peer review of "Vitamin D Status, Cardiovascular Risk Profile, and miRNA-21 Levels in Hypertensive Patients: Results of the HYPODD Study"

_nutrients, 2022, doi:10.3390/nu14132683_

Round 1

Reviewer 1 Report

My specific comments are appended as follows.

1.    It is recommended that authors add data of Serum 25(OH)D3 concentration, blood pressure, biochemical parameters involved in blood pressure regulation and blood lipid profile of participants receiving cholecalciferol or placebo for 2 months, 6 months, 12 months.

2.    “As reported in table 2, at T12, the of systolic blood pressure measured in C-cohort patients is significantly higher compared to this measured in P-cohort study: practically, in C-cohort patients we observed a reduction in systolic blood pressure values significantly higher compared to this observed in P-cohort patients.”  Because there are only 15 people in C-Cohort and only 20 people in P-Cohort, whether the inconsistent factors of “During the study follow-up, 8 C-cohort patients and 10 P-cohort patients did not change the antihypertensive therapy, 6 C-cohort patients and 7 P-cohort patients reduced the dosage and/or the number of taken antihypertensive drugs, and 1 C-cohort patients and 3 P-cohort patients increased the dosage and/or the number of taken antihypertensive drugs” in the text make the hypertension data inaccurate and uncredible.

Author Response

Response to Reviewer 1 Comments

Point 1: It is recommended that authors add data of Serum 25(OH)D concentration, blood pressure,biochemical parameters involved in blood pressure regulation and blood lipid profile ofparticipants receiving cholecalciferol or placebo for 2 months, 6 months, 12 months

Response 1: According to your suggestion, we reported the clinical and biochemical parameters determined at T2, T6, and Tich12 in the two study cohorts in a new table, the table 2. We have also modified the table 3, in whichom the differences from T0 were showed, deleting the T0 column which is now reported in the new table 2.

Point 2: “As reported in table 2, at T12, the of systolic blood pressure measured in C-cohort patientsis significantly higher compared to this measured in P-cohort study: practically, in C-cohortpatients we observed a reduction in systolic blood pressure values significantly highercompared to this observed in P-cohort patients.” Because there are only 15 people in C-Cohortand only 20 people in P-Cohort, whether the inconsistent factors of “During the studyfollow-up, 8 C-cohort patients and 10 P-cohort patients did not change the antihypertensivetherapy, 6 C-cohort patients and 7 P-cohort patients reduced the dosage and/or the numberof taken antihypertensive drugs, and 1 C-cohort patients and 3 P-cohort patients increasedthe dosage and/or the number of taken antihypertensive drugs” in the text make thehypertension data inaccurate and uncredible.

Response 2: According to your suggestion, the sentence in the rResults section was modified. We also discussed this result in the dDiscussion section, underlining its evident limitations (discussion section, page 9, line 348): “This result (the reduction in systolic blood pressure) must be considered with great caution considering the small number of patients enrolled, the primary endpoint of the HYPODD study previously reported, and the changes in the antihypertensive treatment which were adopted by clinicians during the study follow-up according to a predetermined flow-chart [27].

Reviewer 2 Report

Title is concise and relevant and abstraction of the manuscript is adequate. However, bot mÄ°R-21 and vitamin D involve in inflammation and hypertension produce chronic low grade inflammatory burden, I think inflammation could be presented as another keyword.

Background data is too insufficient and must be improved. mÄ°r-21 is associated with inflammatory conditions (https://pubmed.ncbi.nlm.nih.gov/27610006/.). On the other hand, increased serum inflammatory markers have been reported in reduced vitamin D situations (https://pubmed.ncbi.nlm.nih.gov/30266124/). Moreover, hypertension is related with increased serum levels of metabolic and inflammatory predictors (https://pubmed.ncbi.nlm.nih.gov/35142235/). Thus, it is reasonable studying vitamin D and the microRNAs (miRs) system in hypertension. Improve background accordingly, please.

Methodology is clear. Methods expressed properly. Statistics are correct. Authors mentioned that "The distribution of miRs was assessed by Kolmogorov–Smirnov test". Which normality test was applied to other study variables? 

Results presented well. 

Discussion is fair but would be better by two minor revisions. First, I recommend discussing similar works found association between mÄ°R-21 and hypovitaminosis D in chronic metabolic and inflammatory conditions. For example, mÄ°R-21 suppresses inflammation in conditions characterized with increased inflammatory burden (https://pubmed.ncbi.nlm.nih.gov/29884447/). Similarly, low vitamin D levels are associated with type 2 DM (https://pubmed.ncbi.nlm.nih.gov/30758420/). Second, few words about possible clinical benefits of the study results would iprove discussion. Conclusions are justified.

Eleven of the references are older than 10 years and should be replaced with novel works, if possible.

Author Response

Response to Reviewer 1 Comments

Point 1: Title is concise and relevant and abstraction of the manuscript is adequate. However, bot mÄ°R-21and vitamin D involve in inflammation and hypertension produce chronic low grade inflammatory burden, I think inflammation could be presented as another keyword.

Response 1: According to your suggestion, we have added the term Inflammation as new keyword (page 1)

Point 2: Background data is too insufficient and must be improved. mir-21 is associated with inflammatory conditions (https://pubmed.ncbi.nlm.nih.gov/27610006/.). On the other hand, increased serum inflammatory markers have been reported in reduced vitamin D situations (https://pubmed.ncbi.nlm.nih.gov/30266124/). Moreover, hypertension is related with increased serum levels of metabolic and inflammatory predictors(https://pubmed.ncbi.nlm.nih.gov/35142235/). Thus, it is reasonable studying vitamin D and the microRNAs (miRs) system in hypertension. Improve background accordingly, please

Response 2: According to your suggestion, we have implemented the background section of the manuscript adding these sentences (page ): “On the other hand, miR-21 plays a dynamic role in inflammatory responses [24]. The association between vitamin D deficiency and inflammatory status has been well established [25] and hypertension is associated with increased serum levels of metabolic and inflammatory markers [26]. Taking these data into account altogether these data, it is reasonable to studying vitamin D and the miR-21 in patients with hypertension.